# Influence of Amino Acid Substitutions in ApoMb on Different Stages of Unfolding of Amyloids

**DOI:** 10.3390/molecules28237736

**Published:** 2023-11-23

**Authors:** Natalya Katina, Victor Marchenkov, Natalya Ryabova, Nelly Ilyina, Natalia Marchenko, Vitalii Balobanov, Alexey Finkelstein

**Affiliations:** 1Institute of Protein Research RAS, 142290 Pushchino, Russia; march@phys.protres.ru (V.M.); ryabova@phys.protres.ru (N.R.); nelly.ilyina@mail.ru (N.I.); lita@phys.protres.ru (N.M.);; 2Branch of the Shemyakin-Ovchinnikov Institute of Bioorganic Chemistry RAS, 142290 Pushchino, Russia

**Keywords:** apomyoglobin, amyloid stability, unfolding transition, native electrophoresis, hydrophobicity

## Abstract

To date, most research on amyloid aggregation has focused on describing the structure of amyloids and the kinetics of their formation, while the conformational stability of fibrils remains insufficiently explored. The aim of this work was to investigate the effect of amino acid substitutions on the stability of apomyoglobin (ApoMb) amyloids. A study of the amyloid unfolding of ApoMb and its six mutant variants by urea has been carried out. Changes in the structural features of aggregates during unfolding were recorded by far-UV CD and native electrophoresis. It was shown that during the initial stage of denaturation, amyloids’ secondary structure partially unfolds. Then, the fibrils undergo dissociation and form intermediate aggregates weighing approximately 1 MDa, which at the last stage of unfolding decompose into 18 kDa monomeric unfolded molecules. The results of unfolding transitions suggest that the stability of the studied amyloids relative to the intermediate aggregates and of the latter relative to unfolded monomers is higher for ApoMb variants with substitutions that increase the hydrophobicity of the residues. The results presented provide a new insight into the mechanism of stabilization of protein aggregates and can serve as a base for further investigations of the amyloids’ stability.

## 1. Introduction

To function, a protein molecule acquires a unique conformation during the folding process [1,2]. If the protein does not fold into its native state, then it is not able to perform its function and may form aggregates [3,4]. Amyloid fibrils are a specific type of protein aggregates that contribute to the emergence of severe human diseases. Currently, more than 50 diseases are classified as amyloidosis, including Type 2 diabetes, Alzheimer’s and Creutzfeldt–Jakob diseases, and other disorders [5,6]. However, not all amyloids are associated with pathology. Some proteins form functional fibrils that serve a structural role or store biologically active molecules [7,8]. Due to the high stability of amyloid fibrils, several approaches are being developed for their use as high-strength biomaterials [9,10].

Amyloids are long, unbranched protein fibrils that possess a distinctive characteristic: the cross-β-structure, composed of β-strands perpendicular to the aggregate axis, and hydrogen bonds between strands that are parallel to the axis [11,12,13]. There has been a surge in research on amyloid aggregation in the past several years, with an emphasis on investigating the structure of fibrils and the kinetics of their formation [13,14,15,16,17], while the conformational stability of amyloids and the methods of their dissociation remain near to unexplored. Studying the unfolding of amyloids is important for understanding the fundamental principles of protein aggregates’ stabilization. Moreover, knowledge of amyloid unfolding stages and stability is crucial for finding treatments for amyloid diseases. However, investigation of amyloids’ stability is difficult due to aggregates’ insolubility, heterogenicity, and high resistance to denaturation [18,19]. Therefore, to date, fibril disassembly experiments have been carried out for only a few proteins [20,21,22].

A standard approach for studying the stability of monomeric proteins is to probe their equilibrium unfolding by a denaturant [23,24]. In order to investigate the stability of amyloid aggregates, similar approaches can be used. A number of works have been devoted to the study of the stages of amyloids’ unfolding by denaturants [20,21,22]. Using this approach, the contribution of individual residues to the stabilization of prion protein (PrP) amyloids was determined [22]. It was revealed that when insulin and β2-microglobulin fibrils undergo unfolding, intermediate aggregates are produced. The latter do not bind thioflavin T (ThT), but they do retain the characteristic morphology and the secondary structure of amyloids [20].

This work aims to investigate the effect of amino acid substitutions on the stability of an amyloid relative to the unfolded state. ApoMb was chosen as the object for this study. This is an α-helical globular protein comprising 156 amino acid residues. Its structure includes eight helices, designated by letters from A to H [25,26]. The folding of ApoMb occurs through the formation of a molten globule intermediate state, in which the A-, G-, and H-helices retain the native-like topology [27,28]. The precursor of ApoMb amyloids can be either an unfolded or an intermediate (molten globule) state of the protein [29,30,31]. It has also been shown that the amyloid formation propensity of ApoMb increases with the increasing degree of its native state destabilization [31]. Experimental data evidenced that the ApoMb region corresponding to the A-helix is the most amyloidogenic one [30,32]. Therefore, a substantial amount of data have been gathered thus far regarding the folding and amyloid aggregation of ApoMb. This renders this protein a suitable model for investigating the stability of amyloids.

This study is focused on analyzing the stability of amyloids of ApoMb and six of its mutant variants. To assess the stability of the amyloids, they were unfolded by urea. Far-ultraviolet circular dichroism (UV CD) was used to monitor changes in the secondary structure of aggregates as they unfolded. Additionally, blue native PAGE (BN-PAGE) electrophoresis was employed to record variations in the molecular weight of the aggregates. The data obtained allowed us to reveal three stages of amyloids’ disassembly, namely the unfolding of the amyloid secondary structure (i), the dissociation of fibrils to intermediate aggregates (ii), and their subsequent disassembly to monomers (iii). Based on the results of amyloid unfolding transitions, we assessed the stability of amyloids and intermediate aggregates. The hydrophobicity of amino acid residues was found to be a key determinant of stabilization in both of these aggregate types. The contribution of the studied residues to the stabilization of amyloids and intermediate aggregates was estimated.

## 2. Results

### 2.1. At a Temperature of 40 °C and pH 5.5, ApoMb Forms Amyloids

The aim of this work was to study the effect of amino acid substitutions on the stability of ApoMb amyloid fibrils. The hydrophobicity of amino acid residues is widely recognized as a key factor influencing the kinetics of proteins’ amyloid aggregation [5,15]. Therefore, we hypothesized that hydrophobicity may also play an important role in fibrils’ stabilization. We have chosen ApoMb residues from different regions of the protein for substitutions: V10 (A-helix), L115 (G-helix), and M131 (H-helix). For each position, two protein mutant forms with oppositely modified hydrophobicity were obtained. One substitution led to an enhanced hydrophobicity (V10F, L115F, M131W), while the other resulted in a reduced hydrophobicity (V10A, L115A, M131A). In addition, according to our previous results, these mutants differ significantly in the stability of the monomeric protein [31]. Thus, the series of six obtained ApoMb mutant variants was created, providing an appropriate platform for investigating the impact of monomeric protein stability and residue hydrophobicity on amyloid stability.

The aggregates of wild-type (WT) ApoMb and its mutant variants were formed by a 24 h incubation of proteins at 40 °C and pH 5.5. The structural features of amyloid aggregates formed by ApoMb variants were studied previously [31]. As an example, Figure 1 shows the results obtained for the V10F variant, since this protein is characterized by high amyloidogenicity.

Electron microscopy findings have revealed that under the investigated conditions, ApoMb forms twisted fibrillar aggregates that extend up to 100 nm in length (Figure 1a). The FTIR spectra of monomeric ApoMb V10F have one maximum at a wave number of 1650 cm^−1^, where α-helices and a coil absorb (Figure 1b). While in the spectra of aggregates, a second peak at 1620 cm^−1^, characteristic for an amyloid cross-β-structure, appears [33]. Thus, based on the obtained data, it can be concluded that ApoMb and its mutant forms at a temperature of 40 °C and pH 5.5 form amyloid fibrils.

### 2.2. Changes in the Secondary Structure during Amyloids’ Unfolding

The first goal of this work was to choose approaches for studying the amyloids’ stability. The common method for evaluating the stability of monomeric proteins is by measuring their equilibrium unfolding transition through denaturation [23,24]. Similar approaches, with some modification, have also been successfully used to assess amyloid stability [20,21,22]. The unfolding of protein structure at different denaturants’ concentrations was usually monitored by spectral methods, among which ThT fluorescence is most often used for the detection of amyloids. However, it turned out that this method is not applicable for studying the urea unfolding of ApoMb fibrils. Even at a low denaturant concentration, when there is no detectable change in ApoMb aggregates’ secondary structure measured by CD, we observed a decrease in the intensity of ThT fluorescence. This implies that a low urea concentration rather induces dissociation of the ThT bound to the ApoMb amyloid than the destruction of fibrils’ structure. Therefore, in this work, the unfolding of the ApoMb amyloids by urea was studied using the far-UV CD method.

The stability of the monomeric ApoMb and amyloids was evaluated by incubating them at different urea concentrations followed by measuring their CD spectra. Figure 2 compares the unfolding of the monomeric ApoMb V10F variant and its amyloids by urea. The data show that in the absence of a denaturant, the spectrum of the monomer has two distinct minima at wavelengths of 208 nm and 222 nm, which are indicative of α-helical protein structure (Figure 2a). The spectrum of the amyloid has a wide minimum in the range of 210–220 nm, which is a feature of the protein with β-structure. Upon the addition of urea, the unfolding of the protein and aggregates can be clearly observed through a decrease in the molar ellipticity as shown in Figure 2b–d. Due to absorption of urea in the wavelengths up to 210 nm, the CD spectra are presented in the range of 210–250 nm in Figure 2b–d. When the urea concentration reaches 4.8 M, both the monomeric protein and amyloid are fully unfolded. This is evident from the molar ellipticity value of 0–2·10^−3^ deg·cm^2^/dmol in the wavelength range of 215–250 nm (Figure 2d).

The obtained spectra were used to plot the unfolding transitions of the ApoMb monomer and amyloids based on their molar ellipticity at 222 nm (Figure 2e,f). To determine the time required to achieve equilibrium in amyloid solutions, we compared the CD transitions of amyloid unfolding after 1 and 2 days of incubation of aggregates at different urea concentrations. The coincidence of the obtained transitions allowed us to conclude that incubation for 1 day is sufficient to achieve equilibrium, so all subsequent measurements were carried out after this incubation time.

The transition midpoint of monomeric ApoMb V10F is at 2.35 M urea (Figure 2e), while for the amyloid this value shifts towards higher urea concentrations and corresponds to 3.37 M (Figure 2f). These data evidence that the amyloid aggregates are much more stable as compared to the monomer. Thus, using CD to measure amyloid unfolding by urea allows for the evaluation of the fibrils’ secondary structure stability relative to the unfolded protein.

### 2.3. Influence of Amino Acid Substitutions on the Stability of Amyloids’ Secondary Structure

To investigate the influence of amino acid substitutions on the stability of amyloids’ secondary structure, we conducted experiments of the urea unfolding of fibrils of ApoMb and its mutant variants. Figure 3 shows the amyloids’ unfolding transitions obtained by far-UV CD. The values of the midpoints of denaturation transitions ([urea]_1/2_β-U) were calculated from a sigmoidal approximation of the obtained graphs (Table 1). These values serve as a quantitative characteristic of the stability of the amyloids’ secondary structure relative to the unfolded protein.

Our next aim was to determine whether the stability of amyloids’ secondary structure depends on the stability of the monomeric protein. Under the conditions appropriate for amyloid aggregation (pH 5.5, 40 °C), ApoMb molecules adopt either a native or an intermediate (molten globule) state [31]. The populations of these states can be used as a measure of the monomeric protein structure’s stability. The fractions of the native state (f_N_) of ApoMb variants under the studied conditions were calculated in our previous work [31] and listed here (Table 1). Figure 4a shows the midpoints of amyloid denaturation plotted versus the fractions of the native state (f_N_) of the ApoMb variants. The graph’s low correlation coefficient (r = 0.51) indicates that the stability of amyloids’ secondary structure weakly depends on the stability of the monomeric protein.

Then, we hypothesized that the hydrophobicity of amino acid residues may determine the stability of fibrils, as hydrophobic interactions are known to play an important role in the formation of aggregates. However, the correlation between the midpoints of denaturing transitions and the change in the hydrophobicity (from Ref. [34]) resulting from mutations plotted for all studied variants is weak (r = 0.44) (Figure 4b). When analyzing these results, it is important to take into account that various residues may be involved to different extents in the formation of intermolecular interactions. In other words, one residue can be entirely enclosed within the aggregate’s structure, whereas another residue of the same type can form contacts only by part of its surface. To investigate the impact of residues’ hydrophobicity on amyloid stability, it is crucial to compare the stability of fibrils of mutants with substitutions at the same position by different residues. Figure 4c presents this comparison and reveals a general feature for all three positions analyzed. It shows that the secondary structure of amyloids of mutants with substitutions for a more hydrophobic residue are significantly more stable compared to the amyloids of variants with substitutions for a less hydrophobic alanine. These data indicate that increasing in the residue hydrophobicity of ApoMb mutants leads to the stabilization of amyloid fibrils.

### 2.4. Changes in the Molecular Weight of the Aggregates during Amyloids’ Unfolding

The far-UV CD method allowed us to study urea-induced transitions of ApoMb fibrils to unfolded protein and quantify the stability of amyloids’ secondary structure. However, CD data cannot provide information regarding changes in the oligomerization degree of protein aggregates accompanying the unfolding process of the fibrils. For detailed study of amyloid unfolding, it is crucial to determine the molecular weight of protein aggregates applying an appropriate experimental method. To analyze the size of proteins, the gel filtration is widely used. However, its use in the field of amyloid aggregation is often limited due to nonspecific sorption and column clogging by aggregating proteins. Analytical ultracentrifugation experiments are quite time-consuming and complicated by the influence of particles’ morphology on sedimentation [35]. Thus, we have selected blue native PAGE (BN-PAGE) electrophoresis as the method to evaluate the molecular mass of aggregates in the process of ApoMb amyloid unfolding.

For this study, ApoMb fibrils were incubated at various concentrations of urea for 24 h. Then, electrophoresis was carried out without denaturant, with the same amount of protein applied to the slot for all samples. The resulting electrophoresis data for the mutant form V10F are shown in Figure 5a. On the BN-PAGE, the monomeric protein band is clearly visible at the bottom of the gel. With an increase in the urea concentration, an increase in this band intensity is observed, indicating the aggregates’ dissociation. When urea concentration’s range is of 4–6 M, a distinct band appears in the upper section of the gel. This band corresponds to aggregates with a molecular weight of about 1 MDa and provides evidence that amyloid unfolding by urea occurs through the formation of these intermediate aggregates. Also, at high urea concentrations, an additional band with an apparent molecular mass of about 50 kDa is visible, which is supposed to be attributed to the ApoMb dimer. Therefore, it seems that dimers formed during amyloid unfolding are highly stable and do not dissociate even at high concentrations of denaturant.

Although the same amount of the protein was applied to each slot, the total intensities of the lanes vary and show an upward trend as the urea concentration increases. This indicates that the amyloid aggregates, the fraction of which is at a maximum at low urea, are too big to enter the electrophoretic gel and do not contribute to the lane intensity. Figure 5b shows the dependence of the total intensities of the lanes on the urea concentration. When the denaturant concentration reaches 6.2 M, the graph reaches a plateau, indicating the absence of any aggregates that cannot enter into the gel. Thus, at a urea concentration of 8M, the amyloids completely dissociate; therefore, this method can be used to evaluate the stability of ApoMb aggregates as they unfold. From the data obtained, the fraction of large amyloid aggregates $f_{Am(x)}$ can be calculated as:(1)$$f_{Am(x)}=I_{total}-I_{\left(x\right)},$$
where $I_{total}$ is the maximum intensity at the baseline of high (7.5–8 M) urea concentration; x is the urea concentration, (M); $I_{\left(x\right)}$—total intensity of the lane at the urea concentration x.

The fractions of monomers f_M_ and of 1 MDa intermediate aggregates f_Agr_ can be determined by analyzing the intensity of the corresponding bands in the gel. Figure 5c shows the dependence of the fractions of the monomers f_M_, aggregates f_Agr_, and amyloids f_Am_ on the urea concentration. Since the intensity of the dimer band is insignificant, the calculated value f_M_ is the sum of the monomer and dimer bands. The graphs clearly show that decreasing amyloid levels result in an increase in intermediate aggregates, while the disassembly of the latter coincides with the accumulation of the monomeric protein.

The combination of CD and BN-PAGE electrophoresis data analysis provides valuable insights into the stages of ApoMb amyloid unfolding. The midpoint of amyloid secondary structure unfolding transition, measured by CD, is found to be at 3.37 M (Figure 2f and Figure 5d), while the midpoint of the amyloids’ dissociation graph, determined by BN-PAGE electrophoresis, is shifted towards higher urea concentrations and corresponds to 4.71 M (Figure 5c). Therefore, at the first stage of amyloids’ unfolding, the loss of amyloids’ secondary structure occurs (Am_β_→Am_I_). Then, the fibrils dissociate into intermediate aggregates (Am_I_→Agr). At the final stages of amyloid unfolding, intermediate aggregates dissociate into a monomeric protein (Agr→M). The suggested scheme of the ApoMb amyloid unfolding is shown in Figure 5e.

### 2.5. Influence of Amino Acid Substitutions on the Stability of Amyloids Relative to the Intermediate Aggerates and the Stability of Intermediate Aggregates Relative to the Unfolded Monomer

To investigate the impact of amino acid substitutions on the stability of fibrils and intermediate aggregates, the unfolding of amyloids formed by ApoMb mutant variants was studied by BN-PAGE electrophoresis. From the obtained data, the fractions of amyloids f_Am_, intermediate aggregates f_Agr_, and monomers f_M_ versus the urea concentration were calculated using the approach described above (Figure 6).

The dependence of the fractions of ApoMb amyloids on the urea concentration reflects the dissociation of fibrils to intermediate aggregates (Am-Agr) (Figure 6a). From the sigmoidal approximation of the obtained transitions the midpoints [urea]_1/2_Am-Agr were calculated for the ApoMb variants (Table 1). These values were used as a measure of the stability of amyloids (Am) relative to intermediate aggregates (Agr). To determine whether the hydrophobicity of the residues affects the stability of amyloids, the graph of the [urea]_1/2_Am-Agr versus the position of amino acid substitutions was plotted (Figure 7a). The figure shows that fibrils of variants with substitutions that increase hydrophobicity demonstrate a higher stability as compared to those with substitutions for the less hydrophobic alanine.

Moreover, the obtained data also demonstrate that the fractions of amyloids of the ApoMb variants differ before urea addition (points at 0 M urea in Figure 6a). These values represent the fractions of amyloids in the solution once the aggregation process is complete and serve as measures of the ApoMb variants’ aggregation propensities [31]. The transition Am-Agr midpoints calculated from the graphs are independent of the fraction of amyloids prior to the denaturant addition, and they are determined solely by the stability of the fibrils.

Figure 6b represents the fraction of intermediate aggregates as a function of urea concentration. In these graphs, the fraction of aggregates at the urea concentration of 8 M f**_Agr_**_(8M urea)_ is an important characteristic. The presence of a non-zero value indicates the existence of a stable fraction of aggregates that remain nondissociated even in the presence of 8 M urea. The figure shows that for variants with substitutions at position L115 (L115A and L115F), as well as the M131A protein, f_Agr(8M urea)_ is not zero and is in the range 0.15–0.5, evidencing the presence of highly stable aggregates with a high resistance to urea’s action. To reveal the features of these aggregates, we compared the BN-PAGE lanes of amyloids incubated in 8 M urea for the variant V10F without urea-resistant aggregates and L115F with a maximal fraction of aggregates nondissociated by urea (Figure 6d). For the V10F variant, only the monomers and a small fraction of dimers are visible on the BN-PAGE lane. In contrast, for the L115F variant on the BN-PAGE at 8M urea, one can see discrete bands with a molecular mass up to 400 kDa and below. Therefore, urea-resistant aggregates (400 kDa) are distinct from the described earlier intermediate aggregates formed during amyloid unfolding (1 MDa), and represent a specific aggregate type. It can be assumed that the mutations L115A, L115F, and M131A cause destabilization and structural changes in ApoMb, resulting in the formation of relatively small and highly stable aggregates.

The graphs of the increase in the fraction of the monomer with urea concentration indicate the dissociation of intermediate aggregates to monomers (Agr-M) (Figure 6c). From the graph’s approximation by a sigmoid function, the transition midpoints [urea]_1/2_Agr-M were calculated for the ApoMb variants (Table 1). These values are the measure of the stability of aggregates relative to the unfolded monomeric protein. The results show that the stability of intermediate aggregates of the ApoMb variants increases with the increasing hydrophobicity of the substituted amino acid residues (Figure 7b). The impact of the hydrophobicity effect varies significantly depending on the position of the substitution. The greatest differences in the stability of aggregates is observed for mutants with substitutions at position V10: the [urea]_1/2_Agr-M of V10F variant is higher than the [urea]_1/2_Agr-M of V10A by 0.9 M. This indicates the important role of the 10 residue in the formation of interactions in the structure of intermediate aggregates. In contrast, for proteins with mutations at position L115, there is no reliable difference in the value of the transition midpoints. However, the L115F variant forms a greater fraction of urea-resistant aggregates then the L115A protein (Figure 6b). This finding provides further evidence of the higher stability of the L115F aggregates and supports the conclusion that enhancing the hydrophobicity of the residues contributes to the stabilization of ApoMb variants**’** aggregates.

## 3. Discussion

The study of conformations of the polypeptide chain is crucial to understand the mechanism of proteins’ functioning and the development of diseases related to protein misfolding. To date, the folding of globular proteins has been well studied. For many proteins, several conformational states have been described and energy profiles of the folding process have been drawn [23,24,28]. However, the energy landscape of a polypeptide chain includes not only monomeric molecules, but also various types of aggregates [36]. A promising frontier in protein physics lies in the investigation of the structural characteristics and the stability of aggregated conformations.

The aim of our work was to reveal the determinants of ApoMb amyloids’ stability. New approaches for studying the stability of fibrils by monitoring their urea unfolding have been developed. An analysis of the unfolding transitions recorded by CD and BN-PAGE electrophoresis allowed for the determination of the stages of amyloids’ unfolding. As schematized on Figure 5e, at the first step, the secondary structure of protein molecules within amyloid unfolds; then the dissociation of fibrils to intermediate aggregates with a molecular weight of about 1 MDa is observed. At the final stages, these aggregates dissociate into monomeric molecules.

The study of the amyloid stability of ApoMb mutants showed that an increase in hydrophobicity leads to the stabilization of amyloids, as well as intermediate aggregates (Figure 4c and Figure 7). In this work, the stability of the protein variants that differ in the hydrophobicity of the residues at the same position has been estimated. Therefore, by comparing the differences between the stability of amyloids of mutants with substitutions in the same positions by residues that differ in hydrophobicity (∆[urea]_1/2_), we can evaluate the contribution of this residue to the formation of interactions within the aggregates. Table 2 presents these values for both amyloids and intermediate aggregates, calculated using the following formulas:(2)$$∆{\left[urea\right]}_{\frac{1}{2}}\left(V10\right)={\left[urea\right]}_{\frac{1}{2}}\left(V10F\right)-{\left[urea\right]}_{\frac{1}{2}}\left(V10A\right)$$
(3)$$∆{\left[urea\right]}_{\frac{1}{2}}\left(L115\right)={\left[urea\right]}_{\frac{1}{2}}\left(L115F\right)-{\left[urea\right]}_{\frac{1}{2}}\left(L115A\right),$$
(4)$$∆{\left[urea\right]}_{\frac{1}{2}}(M131)={\left[urea\right]}_{\frac{1}{2}}\left(M131W\right)-{\left[urea\right]}_{\frac{1}{2}}\left(M131A\right),$$

Based on the obtained data, it is evident that the value of ∆[urea]_1/2_Am-Agr is the highest (=1 M) for variants with substitutions at the L115 position. Therefore, the 115 residue plays a crucial role in stabilizing amyloids. At the same time, the value of ∆[urea]_1/2_Agr-M for proteins with substitutions at position V10 is significantly higher (=0.9 M), compared to the other positions studied. Therefore, the interactions formed by the 10 residue are the most important for stabilizing the intermediate aggregates. The stability of the secondary structure of amyloids in mutant variants, which vary in hydrophobicity, is nearly identical for all positions. 

## 4. Materials and Methods

### 4.1. Protein Expression and Purification

Plasmids containing the genes of the sperm whale ApoMb mutated forms were obtained using a 7 QuikChange site-directed mutagenesis kit (Stratagene, San Diego, CA, USA) with the plasmid pET17b as a template (a kind gift from Dr. P.E. Wright). ApoMb and its variants were isolated and purified after expression of appropriate plasmids in *Escherichia coli* (*E. coli*) BL21 (DE3) cells as described previously [37].

### 4.2. Amyloids’ Formation

Lyophilized proteins were dissolved in 10 mM sodium phosphate buffer, pH 5.5. Then, the poorly dissolved material was removed by centrifugation using a Beckman 100 ultracentrifuge (Beckman Coulter, Brea, CA, USA) at 90,000 g for 30 min at a temperature of 4 °C. For amyloids’ formation, ApoMb and its variants were incubated for 24 h at a temperature of 40 °C and pH 5.5 at a concentration of 5 mg/mL.

Protein concentration was determined spectrophotometrically. Absorption spectra were measured on a Cary 100 spectrophotometer (Agilent Technologies, Palo Alto, CA, USA) in the 220–450 nm range at a temperature of 25 °C. The extinction coefficients were calculated from the amino acid sequences and taken as $A_{280}^{0.1\%}$ = 0.88 for WT ApoMb and its mutated variants V10A, V10F, L115A, and L115F; the value of $A_{280}^{0.1\%}$= 1.2 was used for the M131W variant [38].

### 4.3. Electron Microscopy

The samples for electron microscopy studies were prepared according to the negative staining method, described earlier [39]. Briefly, a Cooper 400-mesh grid (Electron Microscopy Science, Hatfield, PA, USA) coated with a formvar film was mounted on the protein sample drop (6 µM). After 5 min of adsorption, the grid with the preparation was negatively stained for 2 min with a 1% aqueous solution of uranyl acetate. Micrographs were obtained using a transmission electron microscope JEM 1200 EX (Jeol, Tokyo, Japan) at an 80 kV accelerating voltage.

### 4.4. Fourier Transforms Infrared Spectroscopy (FTIR)

Infrared spectra were recorded at 25 °C on a Fourier IR-spectrometer Nicolet 6700 (Thermo Scientific, Waltham, MA, USA) using a 5 mg/mL protein concentration. Protein samples were sandwiched between two CaF_2_ plates with an optical path length of 5.8 µm. The spectra (averaged from 256 scanning runs) were measured at a resolution of 4 cm**^−^**^1^. The subtraction of buffer spectra and suppression of water steam spectra were conducted using the software OMNIC version 7.4.

### 4.5. Amyloid Unfolding by Urea

For measurements of the urea-unfolding transitions, samples were produced as follows. The monomeric protein and amyloid were subjected to incubation at different urea concentrations, ranging from 0 to 8 M, for a period of 24 h at a temperature of 20 °C. The protein concentration during incubation was 0.5 mg/mL.

### 4.6. Circular Dichroism (CD)

CD spectra measurements were performed on a Chirascan spectropolarimeter (Applied Photophysics, London, UK) with a 0.1 cm path length cuvette. The protein concentration was 0.1 mg/mL. Spectra were recorded in the range 190–250 nm at a temperature of 25 °C. Molar ellipticity ${[\theta }_{\lambda }]$ was calculated according to the formula:(5)$${[\theta }_{\lambda }]=\frac{\theta_{\lambda }\times MRW}{l\times c},$$
where $\theta_{\lambda }$ is the ellipticity value measured at a wavelength of λ, mdeg; MRW is the average residue molecular weight calculated from the amino acid sequence, Da; l is the optical path length, mm; c is the protein concentration, mg/mL. Unfolding transitions were plotted by the molar ellipticity value at a wavelength 222 nm.

### 4.7. Blue Native-PAGE (BN-PAGE) Electrophoresis

Blue native electrophoresis was performed according to [40] with modifications. Briefly, stacking and gradient resolving gels with 5% and 6–21% acrylamide concentrations were used, respectively. The gel-buffer contained 50 mM bis-Tris-HCl, pH 7.0, 0.5 M ε-aminocaproic acid; cathode buffer—15 mM bis-Tris, pH 7.0, 50 mM Tricine, 0.02% Coomassie blue G250; anode buffer—50 mM bis-Tris-HCl, pH 7.0. Protein sample aliquots were supplemented by sample buffer to achieve a final concentration of 30 mM bis-Tris-HCl, pH 7.0, 6% Glycerol, 0.04% Coomassie blue G250. Gels were fixed overnight in 40% Ethanol, 10% acetic acid solution, and stained by colloidal Coomassie blue G250. The bands’ staining intensity was calculated using software Total Lab TL120 (Nonlinear Dynamics Ltd., Newcastle, UK).

## 5. Conclusions

By now, most published studies of amyloid formation describe the kinetics of aggregation and the structure of fibrils, while only a few works are devoted to investigation of the mechanism of amyloid stability. In this work, a new approach was used to study the stability of amyloid aggregates using the ApoMb model as an example. The combined use of far-UV CD and BN-PAGE allowed us to reveal the stages of amyloid unfolding and determine the effect of amino acid substitutions on the stability of both mature amyloids and intermediate aggregates. Based on the experimental results, the contribution of the studied residues to the formation of intermolecular interactions within amyloid fibrils and intermediate aggregates has been assessed. The data obtained provide an insight into understanding the fundamental features of stabilization of protein aggregates, and the presented approaches can be used in further investigations of the stability of the amyloid fibrils formed by different proteins.

## Figures and Tables

**Figure 1 molecules-28-07736-f001:**
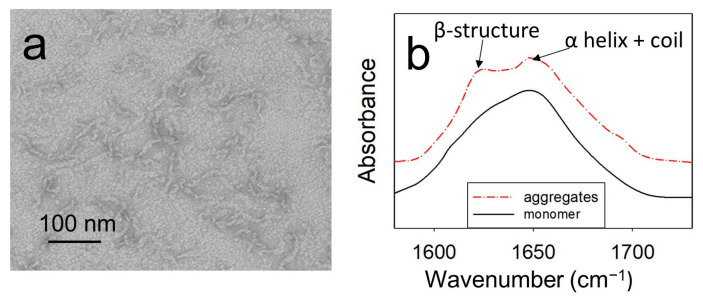
Electron image of aggregates formed by V10F ApoMb (**a**); infrared spectra of aggregates and monomeric protein (**b**).

**Figure 2 molecules-28-07736-f002:**
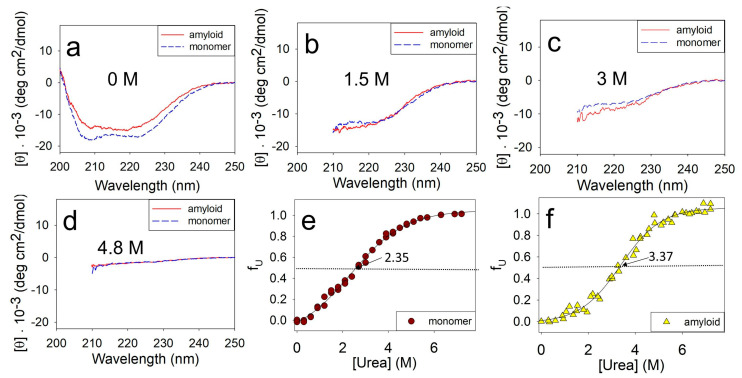
Far-UV CD spectra of the monomer and amyloid of ApoMb V10F at various urea concentrations (**a**–**d**), and normalized unfolding transitions of the monomer (**e**) and amyloid (**f**). Solid lines (**e**,**f**) are the result of a sigmoidal approximation of the experimental data.

**Figure 3 molecules-28-07736-f003:**
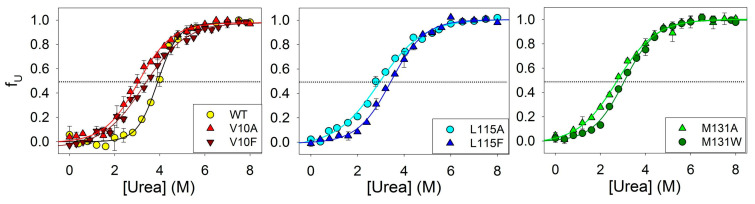
Unfolding of amyloid secondary structure for ApoMb and its mutant variants.

**Figure 4 molecules-28-07736-f004:**
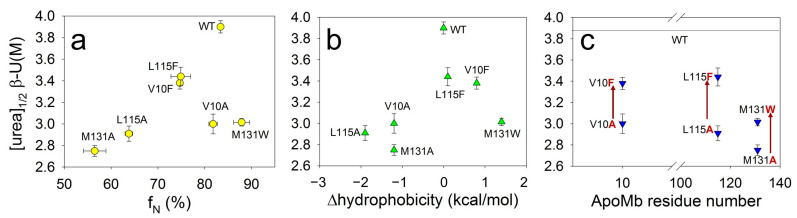
Dependence of midpoints of CD denaturation transitions of ApoMb amyloids ([urea]_1/2_β-U) on the fractions of the native state (f_N_) under conditions appropriate for aggregation (**a**), on the changes in hydrophobicity as a result of mutation (**b**), and on the position of amino acid substitution (**c**).

**Figure 5 molecules-28-07736-f005:**
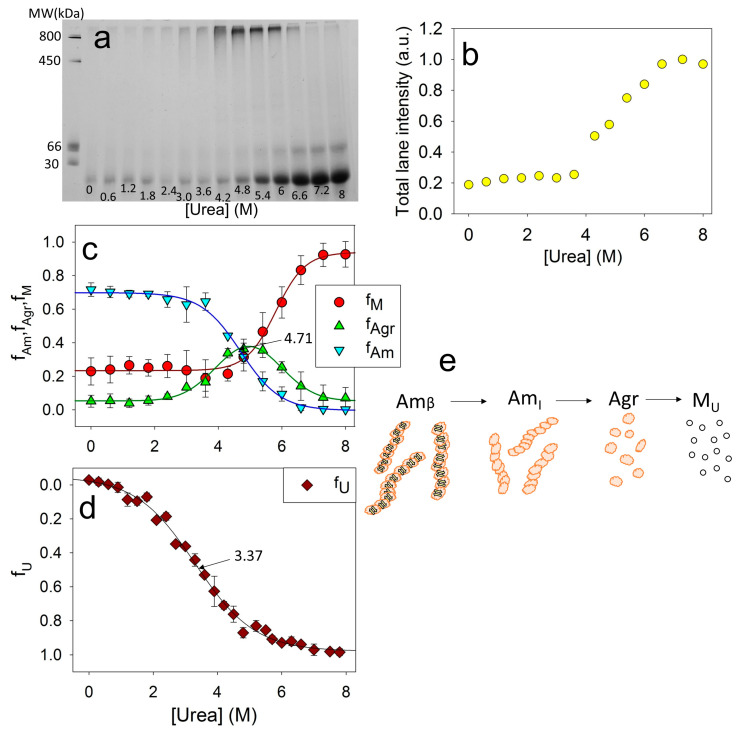
BN-PAGE of ApoMb V10F amyloid solutions after their incubation at various concentrations of urea (**a**); total intensity of the lanes (**b**); fractions of amyloid (f_Am_), intermediate aggregates (f_Agr_), and monomeric protein (f_M_), as a function of urea concentrations (**c**), fraction of unfolded protein (f_U_), as a function of urea concentrations (**d**), supposed model of amyloids’ unfolding, where Am_β_—amyloid with cross-β-structure, Am_I_—intermediate aggregates of molecular weight higher 1 MDa and without cross-β-structure, Agr—intermediate aggregate of 1 MDa, M_U_—unfolded monomer (**e**).

**Figure 6 molecules-28-07736-f006:**
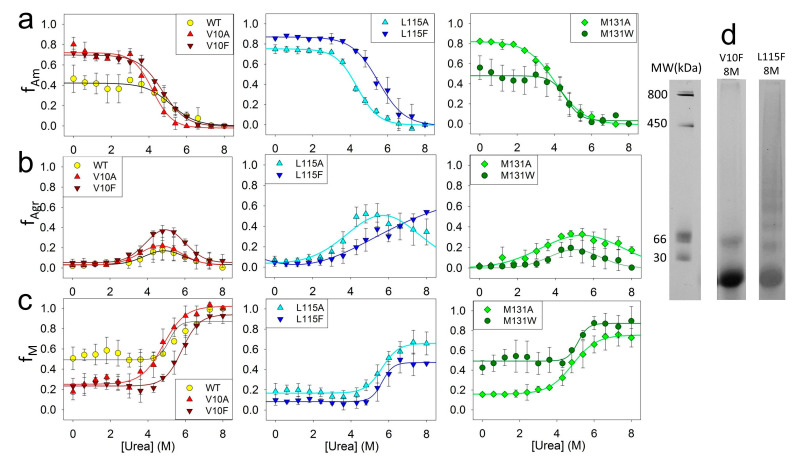
Dependence of the fractions of amyloids (**a**), aggregates (**b**) and monomeric proteins (**c**) for ApoMb and its mutant variants on urea concentration; molecular weights (MW) for protein markers and the BN-PAGE lanes of the V10F and L115F ApoMb variants’ amyloid in 8 M urea (**d**).

**Figure 7 molecules-28-07736-f007:**
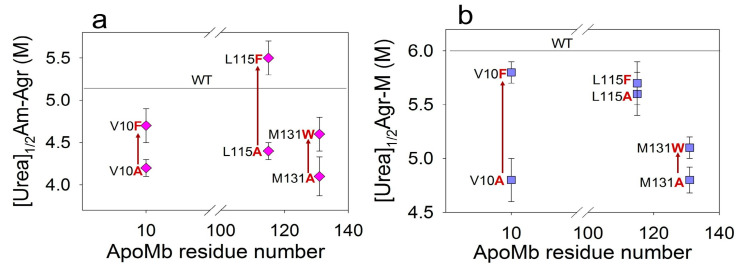
Dependence of midpoints of the amyloids’ (**a**) and intermediate aggregates’ (**b**) dissociation on the positions of the amino acid substitutions.

**Table 1 molecules-28-07736-t001:** Characterization of the stability of monomers and aggregates of ApoMb and its mutant variants.

ApoMb Variant	[urea]_1/2_β-U (M)	f_N_ (%) Data from [31]	∆hydrophobicity (kcal/mol) Octanol/Water Fraction, Data from [34]	[urea]_1/2_Am-Agr (M)	[urea]_1/2_Agr-M (M)
WT	3.9 ± 0.06	83.4 ± 0.6	0	5.2 ± 0.3	6.0 ± 0.4
V10A	3.0 ± 0.09	81.8 ± 0.9	−1.2	4.2 ± 0.1	4.8 ± 0.2
V10F	3.4 ± 0.06	74.7 ± 0.6	0.8	4.7 ± 0.2	5.8 ± 0.1
L115A	2.9 ± 0.07	63.8 ± 0.8	−1.9	4.4 ± 0.1	5.6 ± 0.2
L115F	3.4 ± 0.08	74.9 ± 2.1	0.1	5.5 ± 0.2	5.7 ± 0.2
M131A	2.7± 0.05	56.5 ± 2.4	−1.2	4.1 ± 0.2	4.8 ± 0.1
M131W	3.0 ± 0.04	87.9 ± 1.7	1.4	4.6 ± 0.2	5.1 ± 0.1

[urea]_1/2_β-U (M) is the midpoint of the transition amyloid→unfolded protein, calculated from CD data; [urea]_1/2_Am-Agr (M) is the midpoint of the transition amyloid→intermediate aggregates, calculated from BN-PAGE data; [urea]_1/2_Agr-M (M) is the midpoint of the transition intermediate aggregates→monomer, calculated from BN-PAGE data; f_N_ is the fraction of the ApoMb native state under conditions appropriate for aggregation (pH 5.5; 40 °C), calculated early [31].

**Table 2 molecules-28-07736-t002:** The difference between the amyloids’ and aggregates’ stability for mutant variants with substitutions in the same positions examined.

Position of Substitution	∆[urea]_1/2_β-U (M)	∆[urea]_1/2_Am-Agr (M)	∆[urea]_1/2_Agr-M (M)
V10	0.4 ± 0.2	0.5 ± 0.3	0.9 ± 0.3
L115	0.5 ± 0.2	1.0 ± 0.3	0.2 ± 0.2
M131	0.3 ± 0.2	0.5 ± 0.4	0.3 ± 0.2

## Data Availability

All data are available within the article.

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
