# Peer review of "Influence of Amino Acid Substitutions in ApoMb on Different Stages of Unfolding of Amyloids"

_molecules, 2023, doi:10.3390/molecules28237736_

Round 1

Reviewer 1 Report

Comments and Suggestions for Authors

In this paper, Katina et al investigate the role played by the hydrophobicity of some selected residues in the stability of apoMb amyloids. The goal of exploring protein fibril conformational stability is undoubtedly an interesting, and rather neglected, issue. Based on their results the authors conclude that the  “… stability of studied amyloids relative to the intermediate aggregates and of the latter relative to unfolded monomers is higher for ApoMb variants with substitutions that increase the hydrophobicity of the residues.”

In my opinion this conclusion should be proposed and/or discussed with more caution. In particular, the results of Fig. 2 are discussed as unfolding of amyloid but no evidence is shown to exclude monomer dissociation, or some fibril conformational change.

I am also surprised that midpoint of this “unfolding” transition is only at 3.3 M urea vs 2.3 M urea of apoMb, a rather unstable protein. Fibril unfolding is generally carried out with stronger denaturants and occurs at higher concentrations. This point should also be discussed, together with the reversibility issue.

If I have understood correctly, the amyloid unfolding (Fig 2f and 5d) is actually loss of cross-b secondary structure in an (ordered/disordered?) aggregate AmI. It should be clarified.

Specific points.

It would be appropriate to add the N-state stability of the apoMb mutants as the authors say they have measured.

Show the TEM of the various mutants’ fibrils; are there any morphological (length, curvature, diameter) differences with wt fibrils?

Since hydrophobicity of the mutated residues is a central issue in the manuscript, the choice of the scale used should be indicated.

Efforts should be made to increase the readability of Paragraph 3.5.

Reviewer 2 Report

Comments and Suggestions for Authors

The manuscript by Katina et al., refers to the investigation of the structural stability of fibrils from ApoMb and six of its mutants upon denaturation, describing the dissociation of fibrils to monomeric molecules through intermediate aggregates. Of particular interest are the comparative results derived by native electrophoresis expriments on ApoMb fibrils and its mutants.

This reviewer has a few concerns about the current version of the manuscript:

Lines 103 to 131 are not necessary, Figure 1 included: authors already published similar results in 2011 for V110F mutant, and in 2012 for Met131Ala and Met131Trp mutants; in 2017 for all the V10F, L115F, M131W, V10A, L115A, and M131A mutants. Since are not new findings, there is no need, in the reviewer opinion, to devote a paragraph and a figure in the main text to that. Authors can just refer to published articles.

The readability of 312-326 lines, and of the whole “Discussion” section can be improved.

The “Conclusions” section is missing after the “Materials and Methods” section.

Minor revisions:

In 2. Results section, please correct the numbering of paragraphs

Line 21: “monomeic” →”monomeric”

Line 53: “of monomeric proteins is their..” → “of monomeric proteins is to study their..”

Line 62: “aims to investigate effect..” → “aims to investigate the effect..”

Line 75: “and its six mutant variants”, maybe you intend “and six of its mutant variants”

In Figure 1, “b” label is too close to the curve.

Line 198: “account that that various residues”

Line 206:” residue is significantly” → “residue are significantly”

Line 217: “it’s using in the” → “its using in the”

Line 233: “which are supposed to be” → “which is supposed to be”

Line 270: “(Figure 5d, Figure 2f). While, the” → “(Figure 5d, Figure 2f), while the”

Figure 5.e can be improved in terms of ratio and proportions.

Lines 354-355: “At the first step, the secondary structure of protein molecules within amyloid unfolds (Figure 5e). Then..” → “As schematized in Figure 5e, at the first step, the secondary structure of protein molecules within amyloid unfolds; then..”

line 367: “due to..” → “thanks to..”
